# Epidemiology of Yam Viruses in Guadeloupe: Role of Cropping Practices and Seed-Tuber Supply

**DOI:** 10.3390/v14112366

**Published:** 2022-10-27

**Authors:** Mame Boucar Diouf, Sébastien Guyader, Olyvia Gaspard, Eric Francius, Pierre-Yves Teycheney, Marie Umber

**Affiliations:** 1INRAE, UR 1321 ASTRO, F-97170 Petit-Bourg, Guadeloupe, France; 2CIRAD, UMR AGAP Institut, F-97130 Capesterre Belle-Eau, France; 3UMR AGAP Institut, Institut Agro, Université de Montpellier, CIRAD, INRAE, F-97130 Capesterre Belle-Eau, France; 4CIRAD, UMR PVBMT, F-97410 Saint-Pierre, La Réunion, France; 5UMR PVBMT, Université de la Réunion, F-97410 Saint-Pierre, La Réunion, France

**Keywords:** *Dioscorea* spp., viruses, epidemiology, cropping practices, viral diversity, vertical transmission, weeds, reservoirs

## Abstract

The epidemiology of yam viruses remains largely unexplored. We present a large-scale epidemiological study of yam viruses in Guadeloupe based on the analysis of 1124 leaf samples collected from yams and weeds. We addressed the prevalence of cucumber mosaic virus (CMV), Cordyline virus 1 (CoV1), Dioscorea mosaic associated virus (DMaV), yam asymptomatic virus 1 (YaV1), yam mosaic virus (YMV), yam mild mosaic virus (YMMV), badnaviruses, macluraviruses and potexviruses, and the key epidemiological drivers of these viruses. We provide evidence that several weeds are reservoirs of YMMV and that YMMV isolates infecting weeds cluster together with those infecting yams, pointing to the role of weeds in the epidemiology of YMMV. We report the occurrence of yam chlorotic necrosis virus (YCNV) in Guadeloupe, the introduction of YMMV isolates through the importation of yam tubers, and the absence of vertical transmission of YaV1. We identified specific effects on some cropping practices, such as weed management and the use of chemical pesticides, on the occurrence of a few viruses, but no crop-related factor had a strong or general effect on the overall epidemiology of the targeted viruses. Overall, our work provides insights into the epidemiology of yam viruses that will help design more efficient control strategies.

## 1. Introduction

Yam (*Dioscorea* spp.) is an important staple crop for hundreds of millions of people in the tropics and subtropics, particularly in West Africa, which accounts for 97.8% of the world yam production [1]. In addition to providing an important source of income, yams are valued for their nutritional value and medicinal properties. However, yam cultivation is increasingly threatened by the occurrence of fungal and viral diseases. 

In Guadeloupe, an archipelago of the Lesser Antilles (Eastern Caribbean), the cultivated yam species of *D. alata* and *D. rotundata* account for most of the local yam production, whereas less cultivated species, such as *D. trifida*, *D. esculenta*, and *D. cayenensis*, contribute to a lesser extent. The lack of a seed supply chain results in farmers using non-certified planting material, such as tubers from their previous harvest, bought from other farms, or imported for human consumption purposes, although such a practice is prohibited by law to prevent the spread of pests and diseases [2].

Yams are planted on mounds or ridges, grown on stakes, or crawling on the ground, and weed control is achieved through mulching or by manual or mechanical weeding since the use of herbicides in yam plots has been prohibited in Guadeloupe since 2005. Other crops, such as banana, sugarcane, or pineapple, are often grown in association with yams in the same plots or in nearby plots [2].

Yam remains the most cultivated food crop in Guadeloupe, although its production has decreased sharply from 22,500 tons/year in 1968 to 3000 tons/year in 2006 [1]. Despite increasing briefly to 6300 t in 2016, production has continued to decline, dropping to 2157 t in 2020 [3], with only 227 ha remaining cultivated, from 450 ha in 2016. As a consequence, Guadeloupe is no longer self-sufficient and must import substantial amounts of yam tubers from Costa Rica and Dominica to cover its domestic consumption. The decrease in local production results from a combination of factors, including changes in land use and the impact of pathogens, including viruses. For example, it is assumed that the near disappearance of *D. trifida* from the Caribbean and the Amazon region, from which this species originates, likely results from its high sensitivity to yam mosaic virus (YMV) [4], a potyvirus originating from Africa, whose worldwide spread likely occurred through the exchange of infected yam germplasm [5].

Twenty-five virus species infecting yam are currently recognised by the International Committee on Taxonomy of Viruses (ICTV) [6]. This figure may be underestimated, as the advent of high-throughput sequencing (HTS) and appropriate analytical frameworks are leading to the continuous discovery of new yam-infecting viruses [6]. However, current knowledge about the biology, epidemiology, transmission, symptomatology, and impact of these viruses on production in single or mixed infections remains insufficient in order to develop effective control strategies. Furthermore, the resistance or tolerance of yam-cultivated varieties to viruses remains largely unknown and is still under investigation [7]. In Guadeloupe, nine viruses belonging to the genera *Ampelovirus*, *Badnavirus*, *Macluravirus*, *Potexvirus*, *Potyvirus*, *Sadwavirus* and *Velarivirus* have been reported in the germplasm collection of the Biological Resource Center for Tropical Plants (BRC-TP) and from small-scale surveys of yam plots in farms [4,8,9,10,11,12,13,14]. The molecular diversity of these viruses in Guadeloupe has been characterised [4,9,10,11,13,14], with the exception of that of macluraviruses.

Here, we report a large-scale epidemiological survey of cucumber mosaic virus (CMV), Cordilyne virus 1 (CoV1), Dioscorea mosaic-associated virus (DMaV), yam asymptomatic virus 1 (YaV1), YMV, yam mild mosaic virus (YMMV), badnaviruses, macluraviruses and potexviruses infecting yams in Guadeloupe. We studied the key factors that could potentially affect the occurrence, prevalence and epidemiology of these viruses in the main yam production areas of Guadeloupe, such as cropping practices, the potential of weeds as viral reservoirs, and the role of tuber imports and trade in the introduction and spread of exotic yam virus strains or species. We investigated the vertical (mother-to-daughter plant) transmission of YMMV and YaV1 in *D. trifida*. Our results set a key milestone toward understanding the epidemiology of yam viruses, which is a prerequisite for the design of more efficient control strategies.

## 2. Materials and Methods

### 2.1. Field Surveys and Collection of Leaf Samples

Two surveys were carried out, serving two distinct purposes, respectively. From October 2019 to December 2019, 16 farm plots and 2 experimental plots scattered along the main yam production areas of Guadeloupe (Figure 1A) were surveyed in order to assess the occurrence and prevalence of CMV, CoV1, DMaV, YaV1, YMV, YMMV, macluraviruses and potexviruses in Guadeloupe. For each plot, yam variety, the origin of the seeds, weed management practices and the use of pesticides, if any, were monitored and registered. During this first survey, a total of 780 leaf samples (Figure 1B) were collected from the six yam species encountered (*D. alata*, *D. bulbifera*, *D. cayenensis*, *D. esculenta*, *D. rotundata* and *D. trifida*).

Another survey was carried out in November 2020. It focused on the prevalence of yam viruses in *D. alata*, which is the main cultivated yam species in Guadeloupe. A total of 116 leaf samples were collected from the same two experimental plots that were sampled in 2019 (Figure 1A). These plots were selected because they are typical of the contrasted agro-pedo-climatic conditions encountered in the yam-growing areas of Guadeloupe. The samples were indexed for the same viruses and virus species as above and also for badnaviruses.

For both sampling rounds, the plants were sampled randomly, ensuring that the sampling was representative of the relative abundance of each yam species and variety when they were mixed within plots. One whole leaf of intermediate age (as determined by size, colour, and firmness) was collected per plant, regardless of the presence of symptoms, to avoid bias towards viruses causing more severe symptoms at the expense of viruses causing symptomless infections or mild symptoms.

The samples from the most abundant annual and perennial weed species were collected during both surveys (46 and 80 samples from eight and two plots, respectively) within or at the immediate vicinity (0–5 m away from the edges) of the sampled yam plots, with the aim of searching for potential reservoirs of yam viruses. All samples were stored at −80 °C until processing. A complete and detailed list of the samples collected, including the location and characteristics of the surveyed plots, is provided in Appendix A.

### 2.2. Virus Indexing

Two pieces of 1 × 2 cm were cut from each leaf sample and placed in two separate grinding tubes (MP Biomedicals, Illkirch-Graffenstaden, France). One set of tubes was used for total nucleic acid (TNA) extractions, according to Foissac et al. [15]. TNAs were used for the detection of CMV, CoV1, DMaV, YaV1, YMMV and YMV using virus-specific primers, and macluraviruses and potexviruses using genus-specific primers [12,14]. The second set of tubes was used for the detection of badnaviruses by multiplex-immunocapture-PCR (M-IC-PCR), as described by Umber et al. [10]. For the 2020 survey, an entire leaf of 5 × 5 cm was collected from each sampled plant. A piece of 1 × 2 cm was processed and used as described above. The remaining part of the leaf sample was processed in a grinding bag (Bioreba, Reinach, Switzerland), instead of a grinding tube, for the detection of badnaviruses.

### 2.3. Cloning, Sequencing and Phylogenetic Analyses

The amplification products raised from macluraviruses and YMMV using primer pairs YamMac4F/YamMac5R [12] and YMMV CP 2F/YMMV UTR 1R [16], respectively, were cloned into a pGEM-T Easy^®^ cloning vector (Promega, Charbonnières, France) using the supplier’s protocol. Cloned amplification products (Appendix A) were sequenced by Genewiz (Leipzig, Germany).

Nucleotide sequences were compared using CLUSTALW [17]. Phylogenetic trees were inferred by Maximum Likelihood using IQ-TREE v. 2.2.0 [18], invoking ModelFinder [19] in order to select the best nucleotide substitution model based on the Bayesian information criterion prior to the tree inference. The branch support values were computed using the Ultrafast Bootstrap (UFBoot) procedure in IQ-TREE [20] with 10,000 replicates. Because of the shortness of the sequences used, an additional branch support criterion was used in the form of SH-aLRT [21] with 1000 replicates.

### 2.4. Statistical Analyses of Field Survey Data

The data from the 2019 survey were processed and analysed using the R 4.2.1 statistical software [22]. Virus indexing results were extracted from the complete dataset (Appendix A), used to generate a matrix summarising the number of positive samples for the viruses and virus genera searched in the 18 sampled plots and normalised by scaling with ranked subsampling using the ‘SRS 0.2.3’ package [23]. The crop-related data (area, weed management, use of pesticides and origin of seeds) were encoded into an 18 × 4 matrix using the following unordered factor levels: Basse-Terre/Grande-Terre/Marie-Galante (variable “Area”), manual/mechanical/mulching (“Weed management”), yes/no (“Use of pesticides”) and own/external (“Seed supply”). The putative relationships between the crop-related factors and virus abundances were investigated through generalised linear latent variable models using the ‘gllvm 1.3.3’ package [24]. For this, crop-related variables were chosen as predictors, matrix rows (i.e., the field plots) were added as a random effect, and appropriate distribution (Poisson) and the number of latent variables (3) were selected based on the minimisation of the Bayesian Information Criterion value of the model fit [25] as well as on the graphical observation of the residuals. 

### 2.5. Assessment of Tuber Transmission of Yam Viruses

The vertical transmission of YaV1 and YMMV was monitored in plants originating from fifty-five tubers harvested from ten *D. trifida* plants coinfected by YaV1 and YMMV maintained under field conditions by the BRC-TP. The tubers were planted in pots containing 5 L of a substrate composed of 35% white peat, 37% potting soil, and 28% pozzolan, which were sterilised for 90 min at 90 °C before use. The pots were maintained in an insect-proof greenhouse and monitored three times a week for sprouting and the absence of potential insect vectors. The leaf samples were collected from the plantlets at one and three months after sprouting and used for virus indexing, as described above (Appendix A).

The risk of the introduction of viruses from imported yam tubers was assessed. For this, 44 *D. alata* and two *D. trifida* tubers imported from Costa Rica were purchased from a local supplier and planted in pots that were placed in an insect-proof greenhouse and monitored, as described above. The leaf samples were collected from the plantlets at one and three months after sprouting and used for virus indexing, as described above (Appendix A).

## 3. Results

### 3.1. Prevalence of Viruses Infecting Yam in Guadeloupe

The prevalence of the viruses targeted in this study was estimated from the 780 leaf samples collected in 2019 on six yam species and 18 plots throughout the island (Figure 1; Table 1). Viruses were detected using both species-specific molecular diagnostic tools targeted towards CMV, CoV1, DMaV, YaV1, YMV and YMMV, and genus-specific molecular diagnostic tools targeted towards badnaviruses, macluraviruses and potexviruses.

CMV was not detected in the analysed samples, confirming previous results pointing to the absence of this virus in yam in Guadeloupe [6,12]. The indexing of badnaviruses could not be performed on the samples collected in 2019 because the method used to process these samples in grinding tubes proved incompatible with the detection method. CoV1, DMaV, YaV1, YMV, YMMV, macluraviruses and potexviruses were detected in all three yam production areas (Figure 1B) and varied greatly in prevalence (Figure 2A). YMMV was the most prevalent virus overall (70.6%) in each sampled yam species, except *D. rotundata*, in which YMV was more prevalent (Table 1). YaV1 was the second most prevalent virus (53.6%). The overall prevalence of YMV in the sampled plants was moderate (15.1%) and similar to that of DMaV (15.5%) and CoV1 (14.1%), which have been recently described on yam in Guadeloupe [13,14]. Macluraviruses and potexviruses were the least prevalent viruses in the analysed samples (3.8% and 3.2%, respectively). Despite its high overall prevalence, YaV1 was not detected in the *D. cayenensis* samples. Likewise, macluraviruses were not detected in *D. trifida*, *D. cayenensis*, *D. esculenta* and *D. bulbifera*, whereas potexviruses were not detected in *D. trifida*. DMaV was not detected in *D. trifida* and *D. esculenta*, and CoV1 was not detected in *D. cayenensis* and *D. esculenta* (Figure 2A; Appendix A). The absence of these viruses from *D. bulbifera*, *D. cayenensis*, *D. esculenta* and *D. trifida* may be attributed to the reduced number of samples for these four yam species (Figure 2A).

Virus prevalence was also assessed in 116 *D. alata* samples collected in 2020 in the Godet and Roujol plots, with the addition of badnaviruses following a change in the processing of the samples. Figure 2B shows the prevalence of the virus species and genera targeted by this study. As for the results obtained in 2019 from the samples collected in both plots, CMV, potexviruses and YMV were not detected in any of these samples (Appendix A). The most prevalent viruses were YMMV and YaV1 (Table 1), which were detected in 74.1% and 71.6% of the samples, respectively. The least prevalent viruses were badnaviruses (17.2%) and macluraviruses (11.2%). CoV1 and DMaV showed an identical prevalence of 29.3%.

The majority of the indexed samples collected in 2019 and 2020 were infected by at least one virus (86.5% and 96.6%, respectively) (Figure 3; Appendix A). Double infection was the most represented mixed infection situation (38.7% and 38.8% in the 2019 and 2020 samplings, respectively), outnumbering single infections (23.8% and 20.7% in the 2019 and 2020 samplings, respectively) and predominantly involving YMMV and YaV1. The samples collected in 2019 and 2020 were infected by a maximum of four and six viruses, respectively.

### 3.2. Molecular Diversity of Macluraviruses

The molecular diversity of yam macluraviruses in Guadeloupe was addressed for the first time in this work. To this end, degenerate primers YamMac4F and YamMac5R were used [12] for the amplification of a 292 bp region corresponding to positions 7885–8177 in the genome of yam chlorotic necrosis virus (YCNV-YJish; GenBank accession number MG755240), encompassing the 3′ end of the sequence encoding the coat protein (CP) and the beginning of the 3′ untranslated region (3′-UTR). A total of 28 *D. alata* and 2 *D. rotundata* samples collected in 2019, and 13 *D. alata* samples collected in 2020, respectively, were indexed positive (Appendix A). Nine amplification products originating from *D. alata* and two from *D. rotundata* collected in 2019 were selected because they reflected the diversity of the plots where macluraviruses were detected. Selected amplification products were cloned and sequenced (Appendix A). Phylogenetic analyses performed on the coding part of the sequences (131 nt in size) showed that these sequences were 94.6–100% similar to each other and 81.6–85.5% similar to sequences from YCNV-Kerala [26] and YCNV-YJish [27], respectively (Figure 4). They were also 90.8–93.1% similar to one EST sequence generated from a sample originating from Nigeria [28]. Despite their limited size, the use of two distinct and robust measures of branch support consistently placed the analysed sequences within the YCNV clade, providing the first evidence that YCNV is present in yams in Guadeloupe (Figure 4).

### 3.3. Correlation between Cropping-Related Factors and the Occurrence of Yam Viruses in Guadeloupe

In the first step towards understanding the epidemiology of yam-infecting viruses in Guadeloupe, we investigated the impact of cultural practices and cropping environment on the abundance of yam viruses in the generalised linear latent variable model framework. This analysis allows for characterising the field plots through virus- and crop-related variables and also observing the (co)-occurrence of viruses across the different field plots after integrating the effects of the crop-related variables. The first observation from the ordination (Figure 5) showed that there was no clear partitioning of the field plots. Nonetheless, the first latent variable was strongly associated with the presence of macluraviruses, as well as with field plots Grand Bassin (positive correlation) and Valentin (negative correlation). The second latent variable showed a weaker correlation with CoV1 (positive) and DMaV (negative). The correlation matrix identified two groups of viruses that showed positive within-group correlations (general co-occurrence): YaV1, YMMV and YMV, on one hand, and DMaV, macluraraviruses and potexviruses on the other hand. The viruses from the latter group were not only among the least prevalent overall but had also never been detected in three out of the four field plots of Marie-Galante (Figure 1B; Appendix A).

Comparing the residual covariances from the model fit without and with the crop-related variables revealed that 70% of the covariance in virus abundances is explained by the covariance of the crop-related factors. However, the strong effects of these variables were observed only for CoV1, DMaV, potexviruses and YMV (Figure 6). The factor with the strongest effect was the geographical area, with Marie-Galante being associated with a lower occurrence of DMaV and potexviruses, and to a lesser extent, with that of CoV1. Likewise, DMaV and potexviruses had a lower occurrence in Grande-Terre. Weed management practices affected the occurrence of CoV1, potexviruses and YMV. Mechanical weeding affected potexviruses (positively) and YMV (negatively), while mulching had a positive effect on CoV1 and a negative effect on potexviruses. Lastly, the use of pesticides was associated with a higher occurrence of YMV. The origin of seed tuber supply, on the other hand, had no significant effect on virus occurrence in the context of this work.

### 3.4. Role of Weeds in the Epidemiology of Yam Viruses

To explore the potential role of weeds as viral reservoirs, a total of 126 samples from 35 weed species were collected in and around a random selection of eight of the eighteen sampled yam plots and used for virus indexing (Appendix A). YMMV was detected in eight samples (6.3% of the analysed samples), with one sample of *Cleome viscosa* (family: *Capparidaceae*) originating from a *D. alata* plot (Le Petit Portland), while the remaining seven samples (three samples of *Acalypha indica* L., family: *Euphorbiaceae*; three samples of *Crotalaria retusa,* family: *Fabaceae*; one sample of *Spermacoce latifolia*, family: *Rubiaceae*) originated from another *D. alata* plot (Roujol). No other virus was detected in any of the 126 analysed weed samples (Appendix A).

The amplification products (259 bp) generated from the YMMV-infected weed samples, corresponding to nucleotide positions 9236–9495 of the YMMV-Brazil reference genome (JX470965), were cloned and sequenced. Phylogenetic analyses were performed on the coding part of these sequences, which was 139 nt long and corresponded to the 3′ end of the CP domain. The analyses showed that the sequences amplified from the weeds were 97.1–100% similar to each other and 79.9–100% similar to the sequences of the YMMV isolates sampled in the yams collected in Africa, the Caribbean, India and Brazil (Figure 7). The sequences amplified from the *C. retusa* and *A. indica* samples collected in the Roujol plot were identical, pointing to the existence of plant-to-plant transmission of YMMV. These sequences shared 97.8% identity with the sequence amplified from the *S. latifolia* sample collected from the same plot. Likewise, the sequence amplified from the *C. viscosa* sample collected in Le Petit Portland was 100% identical to a sequence originating from a *D. alata* plant sampled in Guadeloupe in 2002 (isolate YMMV-Guad2, Genbank accession number AF548501; Figure 7). The detection of YMMV in weeds provides evidence that YMMV has a wider host range than previously thought and that several weed species could serve as reservoirs of this virus.

### 3.5. Seed-Tuber Transmission of YMMV and YaV1 in D. trifida

The vertical transmission of YaV1 and YMMV through tubers was investigated in plants originating from the tubers of infected *D. trifida* plants. For this, fifty-five tubers from ten *D. trifida* mother plants co-infected by YMMV and YaV1, the most frequent co-infection situation encountered during the surveys, were planted in individual pots in an insect-proof greenhouse to avoid possible external vector transmission. One and three months after sprouting, the 55 daughter plants were indexed for CMV, CoV1, DMaV, YaV1, YMMV, YMV, badnaviruses, macluraviruses and potexviruses. Only YMMV was detected in 96.4% (53/55) of the daughter plants at one and three months after sprouting, whereas neither YaV1 nor any other virus was ever detected in any of the daughter plants at any time point (Appendix A). These results indicate that virus transmission from infected mother plants to daughter plants through tubers is not systematic and is likely to differ between virus taxa.

### 3.6. Risk of Introduction of Yam Viruses through the Importation of Yam Tubers

We addressed the risk of virus introduction in Guadeloupe through the importation of yam tubers by indexing plants grown from tubers of *D. alata* variety ‘Kabusah’ (44 tubers) and *D. trifida* variety ‘Cousse-couche’ (two tubers) found on local markets and that had been imported from Costa Rica. The plants originating from these tubers were indexed for CMV, CoV1, DMaV, YaV1, YMMV, YMV, macluraviruses and potexviruses, and only YMMV was detected in two of the *D. alata* plants and both *D. trifida* plants (Appendix A). The four PCR products (259 bp) amplified from these YMMV-infected plants were cloned, resulting in five non-redundant sequences: one and three sequences from the first and second *D. trifida* plants, respectively, and one sequence from one of the two *D. alata* plants. Phylogenetic analyses were performed on the coding part of these sequences (139 nt long), as described above. The sequences obtained from the two infected *D. trifida* plants shared 97.8–99.2% identity with a sequence (YMMV-CR1; AF548499) obtained from an infected *D. trifida* plant originating from Costa Rica [8]. There was 84.1–87% identity between the sequences retrieved from imported *D. trifida* and *D. alata*. Interestingly, the only sequence generated from *D. alata* (CR3YMMVDa) was 100% identical to that from a *D. alata* plant of the same variety, ‘Kabusah’, collected in 2019 in Guadeloupe for this study (Kab71YMMVDa; Figure 7). Altogether these results provide strong evidence that YMMV isolates can be introduced in Guadeloupe through the importation of infected tubers intended for human consumption and diverted to planting material.

## 4. Discussion

Yam production has been increasing steadily for the past 20 years in the three main producing countries (Nigeria, Ghana, and Côte d’Ivoire) thanks to constantly improving yields. Meanwhile, yam production was plummeting in some smaller producing countries such as Guadeloupe, which is no longer self-sufficient and relies heavily on imports to meet local demand. Reversing this trend has become of utmost importance to restore food self-sufficiency, as yam remains one of the pillars of food security in Guadeloupe [29,30]. The situation of yam in Guadeloupe is attributable to several factors, including the burden of pests and pathogens such as viruses, whose accumulation in cultivated yam results from the lack of sexual reproduction, which acts as natural sanitation. The design of efficient strategies to control viruses in yams relies on in-depth knowledge of the epidemiology of these viruses, which is currently missing. In this work, we addressed the epidemiological patterns of yam viruses in Guadeloupe.

### 4.1. Prevalence and Diversity of Yam Viruses in Guadeloupe

CMV is one of the most ubiquitous plant viruses, with an estimated host range exceeding 1000 species in 85 families [31]. It is transmitted by several equally ubiquitous aphid species. CMV has been described in yam in several countries in West Africa, although at a very low prevalence (1.6%) [32,33]. CMV is frequently encountered on major crops such as vegetables and banana in Guadeloupe [34,35], where CMV aphid vectors are also widespread [36]. CMV was reported on a single occasion on yam in Guadeloupe, in 1977, through the observation of CMV-like viral particles by electron microscopy [37]. However, this observation was never confirmed since the advent of molecular diagnostics and cannot be ascertained with confidence because the identification method which was used by the authors proved to be unreliable [6]. As we expected, we could not detect CMV in any of the yam and weed samples collected and analysed in this work.

All the other virus species and genera targeted by this work had been previously described on yam in Guadeloupe [4,6,9,11,12,13,14] and in other yam production areas in the world [6,16,26,27,38,39,40,41,42]. Among the 896 yam samples analysed in our work, almost nine out of ten (87.8%) were infected with at least one virus. This result highlights the high level of circulation of viruses in all yam production areas in Guadeloupe that have been known to occur there for decades, such as YMV, YMMV, badnaviruses and potexviruses [4,8,9,10], or that were characterised more recently, such as CoV1, DMaV, YaV1 and macluraviruses [11,12,13,14]. Among these viruses, YMMV was found to be predominant in all sampled yam species except *D. rotundata*, which is known to be mainly infected by YMV [5,16] and DMaV [43]. YaV1 was the second most prevalent virus (55.9%) and was mainly detected in *D. alata* (66.8%), considering that 80.3% of all the collected samples originated from this species. Some associations between yam species and viruses were not found. For instance, macluraviruses were not found in *D. trifida*, *D. esculenta* or *D. bulbifera*, although this result may be biased by a lack of representativeness due to the very low frequency at which these species were encountered during the surveys, coincidentally with the overall low prevalence of macluraviruses in yams in Guadeloupe. Nevertheless, Umber et al. [12] came with similar results when indexing a larger sample of 172 *D. trifida* plants from the BRC-TP germplasm collection. On the contrary, CoV1 was not detected in *D. cayenensis*, while Diouf et al. [14] found it in *D. cayenensis* accessions of the BRC-PT, although at a low prevalence (7.9%; 3/38).

The low prevalence of YMV (9.8%) in the *D. alata* samples collected in 2019 is consistent with that reported by Umber et al. [12] (8.6%) in the *D. alata* accessions of the BRC-TP germplasm collection. The absence of YMV from the *D. alata* samples collected in 2020 might result from a statistical artefact resulting from a smaller sample size (116 vs. 604 *D. alata* samples for 2020 and 2019, respectively). A similar situation may apply to potexviruses, which were detected at a very low prevalence (0.3%) in the *D. alata* samples collected in 2019 and not at all in those collected in 2020.

Although macluraviruses were reported previously in the BRC-TP yam germplasm collection [12], their diversity and taxonomy had never been addressed. The phylogenetic analyses performed during this study strongly suggest that macluraviruses circulating in Guadeloupe belong to the YCNV species clade. YCNV had been previously reported in *D. alata* and *D. nummularia* from Africa, China, India, and Vanuatu [26,27,28] but never in Guadeloupe. Our work provides evidence that *D. rotundata* is also a host species for YCNV and that there is a very high degree of relatedness (94.6–100% identity in a 131 nt sequence) among YCNV isolates from Guadeloupe, regardless of their host yam species and location, suggesting either a more recent introduction into Guadeloupe through a founder effect or a slower rate of evolution as compared to YMMV.

### 4.2. Horizontal and Vertical Transmission of Yam Viruses

Our data confirmed that mixed infections by distinct virus species are the rule rather than the exception in yams, similar to other vegetatively propagated crops [6,11,41,43,44,45,46]. Considering the very high rates of mixed infections that were registered in both the 2019 and 2020 samplings (86.5% and 96.6%, respectively), we hypothesise that both horizontal and vertical transmission might be involved in the epidemiology of yam viruses in Guadeloupe. Vector (horizontal) transmission has been reported only for YMV, YMMV, and Dioscorea bacilliform AL virus (DBALV; *Badnavirus*) so far [47,48,49]. However, vector transmission is common for viruses in genera *Ampelovirus, Badnavirus, Macluravirus, Potexvirus, Potyvirus, Sadwavirus* and *Velarivirus*. It is, therefore, likely that more yam viruses in these genera are also transmitted by vectors, although research effort is needed to provide experimental evidence supporting this hypothesis. In yam, viral infections are primarily attributed to vertical transmission through vegetative propagation, although this mechanism has never been quantitatively assessed. We addressed this issue for the two most prevalent viruses in our sampling, YaV1 and YMMV. Firstly, we showed that seed-tuber transmission of YMMV occurred in plants originating from non-certified imported tubers. This finding raises concerns about the risk of introduction in Guadeloupe of exotic YMMV isolates and possibly other yam viruses through imported tubers. It highlights the need for tighter controls on the sanitary status of imported yam tubers and for enforcing the ban on the use of tubers imported for consumption purposes as planting material. 

Secondly, we showed that YaV1 is not vertically transmitted in *D. trifida* whereas YMMV is transmitted to almost 100% of daughter plants originating from *D. trifida* tubers coinfected by YMMV and YaV1. Our results relate to those of Bertschinger et al. [50], who showed that the levels of tuber transmission in potato differed between viruses. Considering that all the plants used in this study for the vertical transmission assay were coinfected by YMMV and YaV1, we cannot exclude an antagonistic interaction between these two viruses that would limit the accumulation of YaV1 in infected plantlets and reduce its ability to be transmitted through tubers, nor can we rule out the possibility that the titer of YaV1 was too low within infected plants for this virus to be effectively transmitted through tubers. Nevertheless, host genetic factors have been shown to influence the vertical transmission of viruses in other vegetatively propagated crops, such as cassava and potato [50,51,52], and could play a similar role in yam. Despite YaV1 not being tuber-transmitted, this virus was found at a very high prevalence in Guadeloupe, suggesting that it is vector-transmitted. Ampeloviruses are transmitted in the semi-persistent mode by several members in a dozen of genera of pseudococcid mealybugs and soft-scale insects [53], of which two, *Phenacoccus* and *Planococcus*, have been reported on yam in Guadeloupe [54]. Whether mealybug species in these two genera are involved in the transmission of YaV1 in Guadeloupe and elsewhere remains to be investigated. Yet, our results provide clear evidence that vectors are active and very efficient at spreading YaV1 in Guadeloupe.

### 4.3. Role of Mixed Infection in the Etiology of Yam Viral Diseases

Cotransmission of several viruses by the same vector (*e.g.,* aphids), which is common in plants [55], or through vertical transmission, could contribute to the high level of mixed infection that was observed in this work. Mixed infections are known to favour recombination, which plays a key role in virus evolution [56] through the emergence of recombinant strains such as those reported for YMV, YMMV, and yam badnaviruses [4,57,58]. Recombination events can promote the emergence of strains with increased fitness or capable of causing more severe damage to the host plant, as has been reported in the case of PVY in potato [59,60,61,62,63]. Mixed infection by distinct viruses can result in synergistic or antagonist effects. Synergistic effects, which can result in increased symptoms, have been reported in vegetatively propagated crops such as cassava, sweet potato, and potato [64,65,66,67,68] but not yet in yam. Potyviruses have the ability to suppress post-transcriptional gene silencing, interfere with miRNA-guided cleavage and thus promote the replication of other viruses in mixed infections [69,70,71]. If employed by YMMV, which was found in half (53.6%) of the mixed infections registered in this work (Appendix A), these mechanisms could favour the suppression of antiviral response and facilitate mixed infections with other viruses. On the contrary, mixed infections sometimes result in antagonistic interactions, and it can be hypothesised that these antagonisms may prevent mixed infection by too many viruses in a single host, which would corroborate the observations made in this and previous work [66]. The high prevalence of mixed infections in yam is an obstacle to research on the symptomatology and effects of distinct viruses on yam production, as it is impossible to disentangle single effects from mixed infections. Sanitation programs should provide assistance in this regard by generating incompletely sanitised plants that are infected by only one virus [12].

### 4.4. Role of Cropping Practices and Weeds in the Epidemiology of Yam Viruses in Guadeloupe

Cropping practices such as monoculture, intensive production, or the introduction of new resistant cultivars have been shown to affect the incidence of viral diseases in crops [67,72,73]. In this study, we investigated the correlation between the origin of seed tubers, weed management techniques, the use of pesticides, and the distribution and prevalence of yam viruses in Guadeloupe. While some significant correlations were found for some virus/practice combinations, no strong or generic trend could be identified, suggesting a complex interaction between the analysed variables and other factors, such as vector transmission. This could also result from the fact that the yam fields that were surveyed in this work all showed distinct combinations of crop-related variables. Nevertheless, we found that YMMV can infect several annual weed species that are widespread in yam fields in Guadeloupe and that may play a significant role in the epidemiology of YMMV, considering that *Aphis craccivora*, an aphid vector of YMMV [48], is polyphagous and could spread YMMV to a wide range of host plants, including yams. Our results add to the previous identification of several weed reservoirs of YMV in yam fields in Nigeria [74,75]. Conventional wisdom holds that the use of pesticides reduces the incidence of viral diseases by negatively affecting vector transmission. However, our finding that YMV occurrence was positively correlated with the use of pesticides suggests that the relationship between virus occurrence, vector transmission and pesticide use is not straightforward and is likely to result from various interacting factors.

## 5. Conclusions

In this study, our focus was on identifying factors that could potentially affect the occurrence, prevalence, and epidemiology of yam viruses in Guadeloupe. The knowledge gained from our survey allows us to identify some prophylactic strategies for better protection of yam fields, such as the more careful management of weeds and proscribing the use of imported tubers as planting material. Combining these measures with the use of certified virus-free plant material by farmers should benefit yam production in Guadeloupe. However, by limiting our research on virus introductions to viruses already known to Guadeloupe, it is likely that we overlooked other yam viruses not yet reported in Guadeloupe. Future analyses of introduced tubers using HTS technologies will help draw a more detailed picture of the nature of the yam virome introduced into Guadeloupe through imported tubers and compare it to the indigenous yam virome. An important body of work remains to be undertaken, which concerns the role of vector transmission in the epidemiology of yam viruses, but it will first require identifying which vectors may be involved, for such essential information is still lacking at the present time.

## Figures and Tables

**Figure 1 viruses-14-02366-f001:**
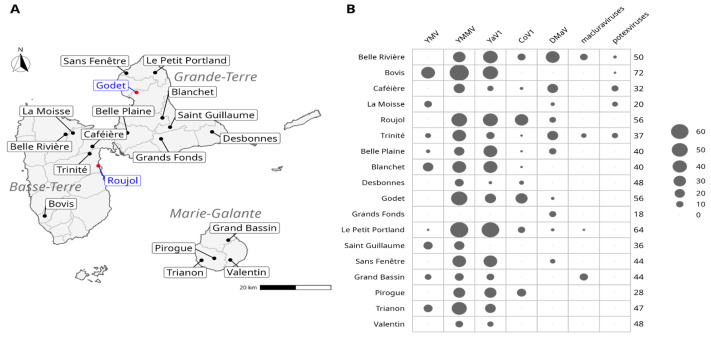
Sampling sites. (**A**) Location of the sampling sites. Sites indicated in blue were sampled both in 2019 and 2020, whereas those indicated in black were sampled in 2019 only. (**B**) Abundance (number of positive samples) of the viruses detected in yams collected from the 18 plots sampled in 2019. The size of the circles is proportional to the number of infected samples. The total number of indexed samples for each site is shown on the right of the table.

**Figure 2 viruses-14-02366-f002:**
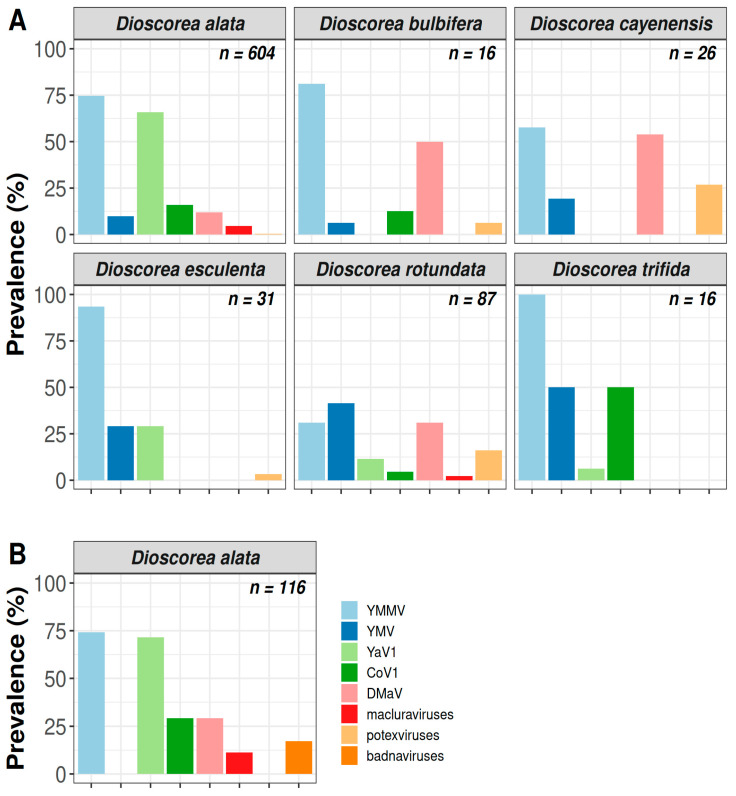
Prevalence of CoV1, DMaV, YaV1, YMV, YMMV, macluraviruses, potexviruses and badnaviruses in yams in Guadeloupe. (**A**) Prevalence of CoV1, DMaV, YaV1, YMV, YMMV, macluraviruses and potexviruses in samples from six yam species collected from 18 yam plots in 2019; (**B**) Prevalence of CoV1, DMaV, YaV1, YMV, YMMV, macluraviruses, potexviruses and badnaviruses in *D. alata* samples collected in 2020 from the Godet and Roujol yam plots. The total number of indexed samples for each yam species is provided on the top right corner of each panel.

**Figure 3 viruses-14-02366-f003:**
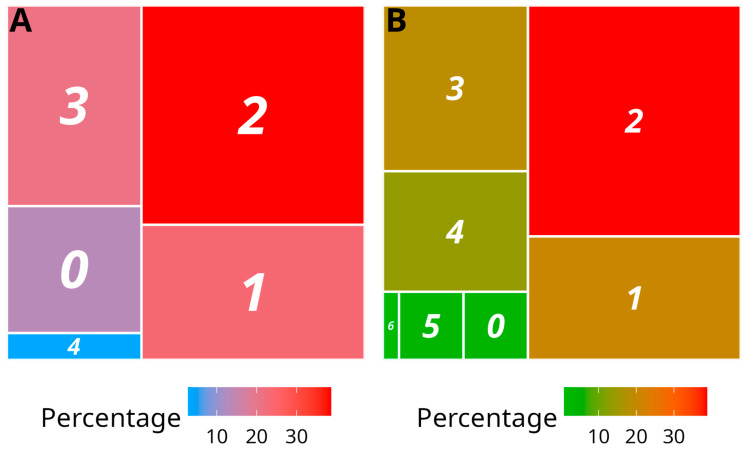
Treemap representation of the infection status of analysed yam samples. (**A**) Samples collected in 2019 from 18 plots and 6 yam species, indexed for CoV1, DMaV, YaV1, YMV, YMMV, macluraviruses and potexviruses. (**B**) Samples collected in 2020 from 2 plots (Godet and Roujol) and one yam species, *D. alata*, indexed for CoV1, DMaV, YaV1, YMV, YMMV, badnaviruses, macluraviruses and potexviruses. The number of distinct viruses detected is shown in the colour boxes, ranging from 0 (no virus detected) to 6 (sextuple infection). The size and colour of the boxes reflect the percentage of samples in each category. A colour scale is provided under the treemaps.

**Figure 4 viruses-14-02366-f004:**
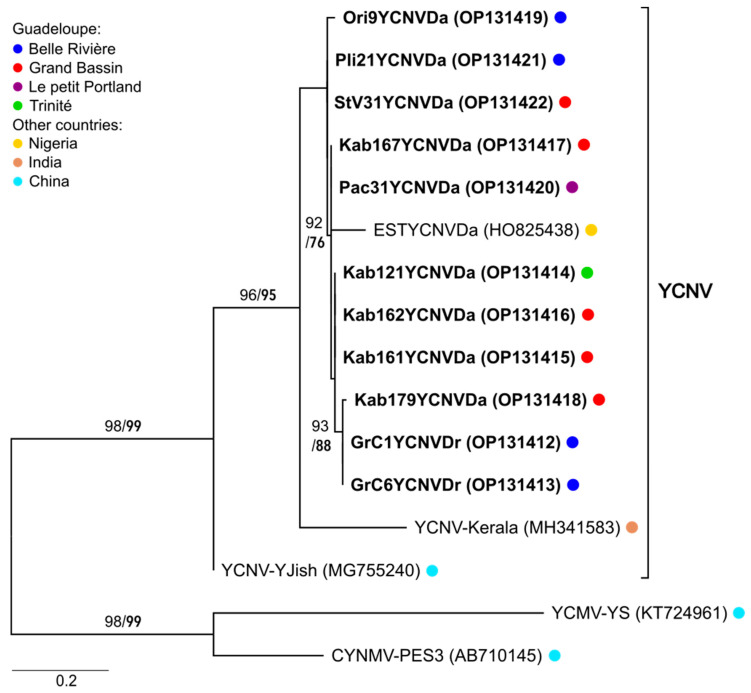
Maximum Likelihood phylogenetic tree showing the relationships between the sequences obtained in this study (in bold) and those of previously characterised YCNV isolates. Phylogenetic analyses were performed on the coding part (131 nt) of amplified sequences corresponding to nucleotide positions 7903–8033 in the genome of YCNV-YJish. An arbitrary root was positioned at the branch separating YCNV from other macluraviruses infecting yam (yam chlorotic mosaic virus and Chinese yam necrotic mosaic virus). The key to the coloured dots referring to the location of the samples from which sequences originate is shown in the box on the left of the figure. Numbers above the branches show SH-aLRT (in normal font) and UFBoot (in bold) give the branch support values expressed in percent of the sampled trees.

**Figure 5 viruses-14-02366-f005:**
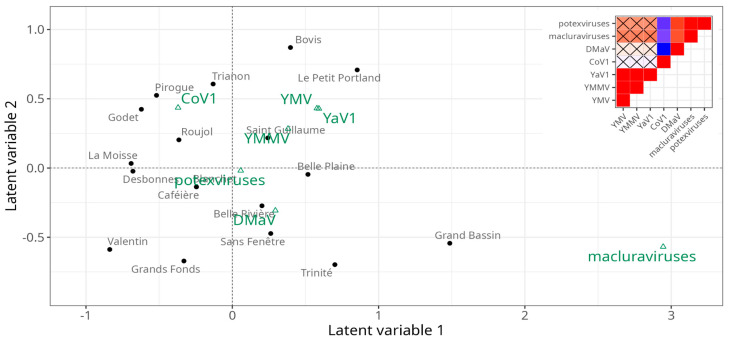
Ordination plot showing the position of yam viruses (green) and sampled field plots (dark grey) relative to the two first latent variables from the gllvm analysis. Elements that are close together on the ordination plot are statistically associated with each other. The diagram in the upper right corner shows the correlation matrix of the viruses, with blue representing negative correlation and red positive correlation between pairs of viruses. Non-significant correlations at the *p* = 0.05 threshold are marked with a cross sign.

**Figure 6 viruses-14-02366-f006:**
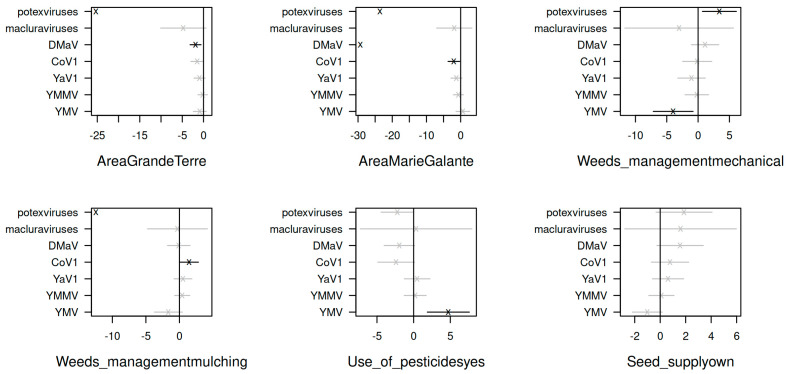
Plots of the coefficients (cross signs) and their standard errors (bars) from the gllvm analysis show the effects of crop-related variables on the standardised abundance of yam viruses in Guadeloupe. Significant and non-significant effects are shown in black and grey, respectively.

**Figure 7 viruses-14-02366-f007:**
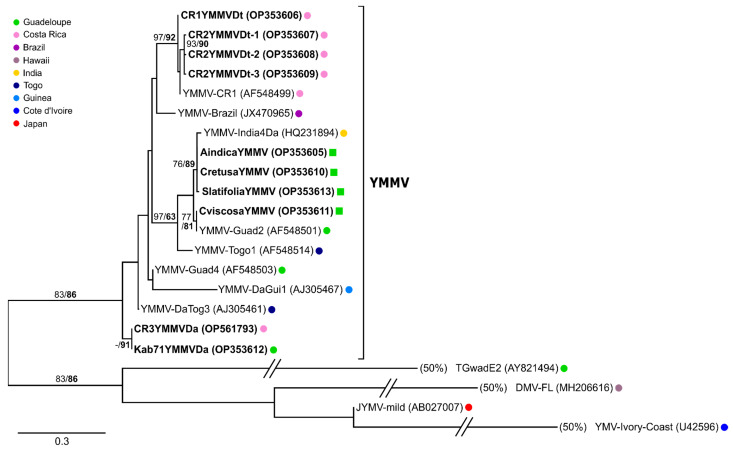
Maximum Likelihood phylogenetic tree showing the relationships between YMMV sequences generated in this study (in bold) and similar sequences from previously characterised YMMV isolates. Phylogenetic analyses were performed on the coding part (139 nt) of the amplified sequences, corresponding to nucleotide positions 9257–9395 in the genome of YMMV-Brazil. An arbitrary root was positioned at the branch separating YMMV from other potyviruses infecting yam. Branches leading to the three sequences at the bottom of the tree were shortened to 50% of their original length in order to fit in the figure. Sequences obtained from yam are represented by coloured dots and those from weeds by coloured squares. The key to the colour dots referring to the location of the samples from which sequences originated is shown in the box on the left of the tree. SH-aLRT (in normal font) and UFBoot (in bold) values shown above branches are expressed in percent of the sampled trees.

**Table 1 viruses-14-02366-t001:** Results of the prevalence survey of yam viruses in Guadeloupe. In addition to the overall number of samples indexed for each yam species, the number (N) and percentage (%) of infected samples are given for each yam species/virus combination.

Yam Species	Number of Indexed Samples	CMV		CoV1		DMaV		YaV1		YMV		YMMV		Macluraviruses		Potexviruses		Badnaviruses	
		N	%	N	%	N	%	N	%	N	%	N	%	N	%	N	%	N	%
**2019**																			
*Dioscorea alata*	604	0	0	96	15.9	72	11.9	398	65.9	59	9.8	451	74.7	28	4.6	2	0.3	NA	NA
*Dioscorea bulbifera*	16	0	0	2	12.5	8	50.0	0	0.0	1	6.3	13	81.3	0	0.0	1	6.3	NA	NA
*Dioscorea cayenensis*	26	0	0	0	0.0	14	53.8	0	0.0	5	19.2	15	57.7	0	0.0	7	26.9	NA	NA
*Dioscorea esculenta*	31	0	0	0	0.0	0	0.0	9	29.0	9	29.0	29	93.5	0	0.0	1	3.2	NA	NA
*Dioscorea rotundata*	87	0	0	4	4.6	27	31.0	10	11.5	36	41.4	27	31.0	2	5.7	14	16.1	NA	NA
*Dioscorea trifida*	16	0	0	8	50.0	0	0.0	1	6.3	8	50.0	16	100.0	0	0.0	0	0.0	NA	NA
Sub total	780	0	0	110	14.1	121	15.5	418	53.6	118	15.1	551	70.6	30	3.8	25	3.2	-	-
**2020**																			
*Dioscorea alata*	116	0	0	34	29.3	34	29.3	83	71.6	0	0.0	86	74.1	13	11.2	0	0.0	20	17.2
**Total**	**896**	**0**	**0**	**144**	**16.1**	**155**	**17.3**	**501**	**55.9**	**118**	**13.2**	**637**	**71.1**	**43**	**4.8**	**25**	**2.8**	**-**	**-**

## Data Availability

The data presented in this study are available in Appendix A. The full dataset is openly available in Recherche Data Gouv at https://doi.org/10.57745/KD1GEB, UNF:6:l34HgP31WUFIMsliouqJtQ==. The nucleotide sequences reported in this work have been deposited in the GenBank database under accession numbers OP131412—OP131422, OP353605—OP353613 and OP561793.

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
