# Peer review of "Epidemiology of Yam Viruses in Guadeloupe: Role of Cropping Practices and Seed-Tuber Supply"

_viruses, 2022, doi:10.3390/v14112366_

Round 1
Reviewer 1 Report
The manuscript presented by Mame Boucar Diouf et al. analyzed a large-scale epidemiological of yam viruses in Guadeloupe, based on the analysis of 1124 leaf samples collected from yams and weeds. They also identified specific effects on some cropping practices such as weed management and the use of chemical pesticides on the occurrence of a few viruses. The authors mentioned that the diseased samples were collected from 2019 to 2020. However, it is regrettable that the data of these two years have poor correspondence, and it is impossible to judge the epidemic trend of these viruses. Generally, the manuscript presented and well analyzed quite a lot of data and written well. I would like to offer the following suggestions with the aim to improve the manuscript.
Comments:
Line 26 "spp" should be "spp."
Lines 39 "tubers" should be deleted
Line 76: "Two surveys were carried out, serving two distinct purposes" should be followed by ", respectively".
Line 203: "were" should be "where".
Line 206: "YCNV-Yjish" should be consistent with the words "YCNV-Y Jish" in the Figure 4.
Line 248: The authors mentioned that "the use of pesticides was associated with a higher occurrence of YMV". As we know, in general, the use of pesticides will reduce the transmission of vectors, thereby reducing the incidence of viral diseases. The author draws the opposite conclusion here. How should we understand this conclusion?
Line 249: As we know, most yam viruses can be transmitted through germplasm exchange. The types of virus carried by seed tubers from different sources may vary. How do you explain "the origin of seed tuber supply, on the other hand, had no significant effect on virus occurrence "?
In the "Materials and Methods", the sampling method should be clearly stated. Is it random sampling or only symptomatic plants? It is better to show the symptoms of yams and weeds infected with viruses.
The author is suggested to discuss the following issues in the "Discussion" section:
The data in Table 1 shows that YMV and potexviruses were detected in 2019, but not in 2020? What is the possible reason?
YMMV was found to be predominant in all sampled yam specie, and YaVl was the second most prevalent virus. What are the possible reasons for the prevalence of these two viruses? How much does it affect the yield and quality of yams?
The author found that YaV1 could not spread vertically through tubers? However, its incidence rate in the field is high. What is the possible reason?
Author Response
- The manuscript presented by Mame Boucar Diouf et al. analyzed a large-scale epidemiological of yam viruses in Guadeloupe, based on the analysis of 1124 leaf samples collected from yams and weeds. They also identified specific effects on some cropping practices such as weed management and the use of chemical pesticides on the occurrence of a few viruses. The authors mentioned that the diseased samples were collected from 2019 to 2020. However, it is regrettable that the data of these two years have poor correspondence, and it is impossible to judge the epidemic trend of these viruses. Generally, the manuscript presented and well analyzed quite a lot of data and written well. I would like to offer the following suggestions with the aim to improve the manuscript.
We agree with reviewer #1 that our data do not allow to judge the epidemic trend of the viruses studied. We hypothesise that such a trend cannot be observed because (i) the viruses targeted by our study are well established in Guadeloupe, and (ii) we have no control on the cropping practices implemented by farmers in the surveyed plots, some of which having an impact on the epidemiology of yam viruses in Guadeloupe.
- Line 26 "spp" should be "spp."
Correction has been made in the revised manuscript
- Lines 39 "tubers" should be deleted
Correction has been made in the revised manuscript
- Line 76: "Two surveys were carried out, serving two distinct purposes" should be followed by ", respectively".
Correction has been made in the revised manuscript
- Line 203: "were" should be "where".
Correction has been made in the revised manuscript
- Line 206: "YCNV-Yjish" should be consistent with the words "YCNV-Y Jish" in the Figure 4.
Correction has been made in the revised manuscript
- Line 248: The authors mentioned that "the use of pesticides was associated with a higher occurrence of YMV". As we know, in general, the use of pesticides will reduce the transmission of vectors, thereby reducing the incidence of viral diseases. The author draws the opposite conclusion here. How should we understand this conclusion?
We think that the relationship between the use pesticides and virus transmission by vectors is not straightforward. Different types of pesticides (herbicides, insecticides...) may affect the transmission of viruses at different levels: herbicides may reduce herbaceous reservoir populations, whereas insecticides may reduce the populations of insect vectors. The conclusion we draw for YMV results from the data analysis reported in this paper and is not an extrapolation from our side: higher occurrence of YMV was associated with the use of pesticides. The cause of this is unknown and can only be speculated.
- Line 249: As we know, most yam viruses can be transmitted through germplasm exchange. The types of virus carried by seed tubers from different sources may vary. How do you explain "the origin of seed tuber supply, on the other hand, had no significant effect on virus occurrence "?
The exchange of infected germplasm or tubers can affect the occurrence and/or prevalence of viruses when these viruses are still emerging, i.e invading the territory. Once viruses have been established for a significantly long period, as is the case for at least some of the viruses targeted by this paper, the exchange of infected tubers within Guadeloupe is not expected to have a significant impact on the distribution of viruses.
- In the "Materials and Methods", the sampling method should be clearly stated. Is it random sampling or only symptomatic plants? It is better to show the symptoms of yams and weeds infected with viruses.
The description of the sampling method has been modified to provide more detail (lines 84 and 90—94). As specified in the manuscript, we sampled at random to avoid bias towards viruses causing more severe symptoms at the expense of viruses causing symptomless infections. Indeed, some of the targeted viruses are known to cause only mild or no symptoms at all. In addition, photographs of typical viral symptoms on yam are shown in the review paper of Diouf et al. (ref. #6). To the best of our knowledge, no such photograph of symptoms caused on weeds by yam viruses exist.
The author is suggested to discuss the following issues in the "Discussion" section:
- The data in Table 1 shows that YMV and potexviruses were detected in 2019, but not in 2020? What is the possible reason?
As explained in the Materials and Methods section, the data collected in 2020 originated from only two plots (Godet and Roujol) which were planted with D. alata. This yam species has already been reported as less prone to infection by YMV and potexviruses (Umber et al. 2020, ref. #12). Since YMV was found to occur at a low prevalence in D. alata in the 2019 sampling, the absence of YMV in the 2020 sampling is likely to be a statistical artefact resulting from a smaller sample size (116 vs. 604 D. alata samples for 2020 and 2019, respectively). Likewise, potexviruses were detected only in two of the 604 D. alata samples collected in 2019, therefore it is not surprising that they were not detected in the samples collected for the very same reasons. Furthermore, neither YMV nor potexviruses were detected in Godet and Roujol plots in 2019. A short paragraph has been added in the Discussion (lines 356—360).
- YMMV was found to be predominant in all sampled yam specie, and YaVl was the second most prevalent virus. What are the possible reasons for the prevalence of these two viruses? How much does it affect the yield and quality of yams?
The high prevalence of YMMV appears to be related to a combination of several factors: the high efficiency of transmission through tubers (and also by vectors as is often the case with non-persistently vector-transmitted viruses), the existence of several reservoir plants and a possible flow of introduction through the importation of tubers. The question regarding YaV1 is addressed below.
The impact of yam viruses on tuber yield and quality still needs to be addressed, this issue has been raised in a recent review cited in the manuscript (Diouf et al. 2022, ref #6).
- The author found that YaV1 could not spread vertically through tubers? However, its incidence rate in the field is high. What is the possible reason?
This suggests that the prevalence of YaV1 is related to highly efficient vector transmission as mentioned in lines 393—400. We slightly rephrased this paragraph to make our point clearer.
Reviewer 2 Report
This MS presents the results concerning the epidemiology and abundance of the main known yam viruses in Guadeloupe. The MS is well written and the results are well presented. The authors should add some information that are missing concerning their sampling procedure (e.g. number of leaves/plant, observation of symptoms, young leaves etc). Furthermore, they state that a modification of the grinding procedure was done in order to detect badnaviruses but there is no information about this modification. Finally, although there is a paragraph (4.3) about the correlation of mixed infections with the disease severity etc. in the discussion section there is no information if this was the situation observed in Guadeloupe.
Some minor comments and suggestions concerning English language can be found in the attached file

Author Response
- This MS presents the results concerning the epidemiology and abundance of the main known yam viruses in Guadeloupe. The MS is well written and the results are well presented. The authors should add some information that are missing concerning their sampling procedure (e.g. number of leaves/plant, observation of symptoms, young leaves etc).
Information about sampling has been added in the Materials and Methods section (lines 84 and 90—94).
- Furthermore, they state that a modification of the grinding procedure was done in order to detect badnaviruses but there is no information about this modification.
This modification is detailed in the Materials and Methods section (lines 110-113). During the 2019 survey, only small leaf pieces were processed in grinding tubes and this proved inadequate for detecting badnaviruses. During the 2020 survey, larger leaf pieces were processed in grinding bags, allowing the successful detection of badnaviruses.
- Finally, although there is a paragraph (4.3) about the correlation of mixed infections with the disease severity etc. in the discussion section there is no information if this was the situation observed in Guadeloupe.
As stated more clearly in lines 90—94, we purposely performed random sampling, regardless of the presence of symptoms on sampled plants to avoid any bias (see comment to a remark of reviewer #1 above). Thus, we cannot correlate single or mixed infections with disease severity in Guadeloupe. However, we feel that opening discussion on the potential effects of mixed infections on disease severity is relevant in the context of our study, to highlight the lack of knowledge about the symptoms associated with single and mixed infections, and the difficulties to correlate symptoms to viruses because single infections are very rare in yam. A short text has been added (lines 416—419).
- Some minor comments and suggestions concerning English language can be found in the attached file.
Corrections have been made accordingly throughout the manuscript.